# Interfacility transfer of pregnant women using publicly funded emergency call centre-based ambulance services: a cross-sectional analysis of service logs from five states in India

Samiksha Singh,[1] Pat Doyle,[2] Oona MR Campbell,[3] Laura Oakley,[2] GV Ramana Rao,[4] GVS Murthy[1,5]

► Prepublication history and additional material are available. To view these files please visit the journal online (http://dx.doi.org/10.1136/bmjopen-2016-015077).

For numbered affiliations see end of article.

**Correspondence to**
Dr Samiksha Singh;
samiksha.singh@iiphh.org

## ABSTRACT

**Objective** To estimate the proportion of interfacility transfers (IFTs) transported by '108' ambulances and to compare the characteristics of the IFTs and non-IFTs to understand the pattern of use of '108' services for pregnant women in India.

**Design** A cross-sectional analysis of '108' ambulance records from five states for the period April 2013 to March 2014. Data were obtained from the call centre database for the pregnant women, who called '108'.

**Main outcomes** Proportion of all pregnancies and institutional deliveries in the population who were transported by '108', both overall and for IFT. Characteristics of the women transported; obstetric emergencies, the distances travelled and the time taken for both IFT and non-IFT.

**Results** The '108' ambulances transported 6 08 559 pregnant women, of whom 34 993 were IFTs (5.8%) in the five states. We estimated that '108' transferred 16.5% of all pregnancies and 20.8% of institutional deliveries. Only 1.2% of all institutional deliveries in the population were transported by '108' for IFTs—lowest 0.6% in Gujarat and highest 3.0% in Himachal Pradesh. Of all '108' IFTs, only 8.4% had any pregnancy complication. For all states combined, on adjusted analysis, IFTs were more likely than non-IFTs to be for older and younger women or from urban areas, and less likely to be for women from high-priority districts, from backward or scheduled castes, or women below the poverty line. Obstetric emergencies were more than twice as likely to be IFTs as pregnant women without obstetric emergencies (OR=2.18, 95% CI 2.09 to 2.27). There was considerable variation across states.

**Conclusion** Only 6% institutional deliveries made use of the '108' ambulance for IFTs in India. The vast majority did not have any complication or emergency. The '108' service may need to consider strategies to prioritise the transfer of women with obstetric emergency and those requiring IFT, over uncomplicated non-IFT.

### Strengths and limitations of this study

► This study is the first to assess the role of the '108' ambulance service— the largest provider of the free emergency medical services in India—for interfacility transfers (IFTs) of pregnant women.
► We assessed the characteristics of pregnant women who were transported by '108' as IFTs and compared them with those who were transported by '108' as non-IFTs, for five states.
► We did extensive cleaning and management of data to drive appropriate information.
► '108' service did not record information on postpartum obstetric emergency separately thus these could not be estimated.
► Diagnosis of obstetric emergency may be inaccurate and subject to interobserver bias.
► The '108' database mostly did not record data for treatment given en route, and doctors' notes on IFTs, thus we could not study these data.
► Some population estimates are based on assumptions and may not be accurate.
► We had large proportions of missing information on social and economic status from two states. We did a complete analysis and sensitivity analysis to deal with missingness.

India had an estimated maternal mortality ratio (MMR) of 167/100 000 live births and an early neonatal mortality rate of 28/1000 live births in 2013.[2] The country accounts for 17% (50 000) of global maternal deaths per year[3] and 26% (696 000) of global neonatal deaths.[4] Many maternal deaths occur during transit to health facilities.[5–7]

Interfacility transfers (IFTs) for pregnant women are crucial, especially in resource-poor countries where most peripheral health facilities provide only uncomplicated birthing or basic emergency obstetric care.[8 9] About 14%–36% of women delivering in facilities

## BACKGROUND

India had an estimated 83% women delivering in health facilities in 2013.[1] Despite this high proportion of institutional births,

are referred from lower-level to higher-level facilities.[10–13] As IFT is more likely to be due to referral for high-risk pregnancies or complications during pregnancy, childbirth or post-partum, it can play a pivotal role in reduction of maternal morbidity and mortality.[14 15]

Ensuring an uneventful IFT is part of good healthcare provision at the referring facility, and will reduce delays in access to appropriate healthcare (delay type 2) and delays in getting appropriate care after reaching a health facility (delay type 3).[16 17] A good process will include prompt arrangement of transport, en route stabilisation, communication with the referral facility to prepare them for the patient and appropriate hand over on arrival. IFT is thus a complex coordinated effort made by the referring healthcare provider, the en route attendant, the receiver at the referral health facility and the referral transport system.[16] Several successful interventions have been reported in resource-poor countries to improve referral transportation, but only a few mention IFT.[18 19] Referral transport for IFT may be (a) an ambulance based at the referring facility, (b) an ambulance called from a referral facility, (c) an independent ambulance service, (d) other subsidised public or commercial transport or (e) personal or commercial (non-subsidised) transport.[16 20]

India, currently, does not have any structured interfacility referral and transportation protocols for pregnant women (desk review and personal communication with Maternal health specialist, Government of India). In India, most peripheral health facilities do not have functional ambulances of their own and if they do, they are not available round the clock.[14 21 22] Thus, IFT for pregnant women depends on other referral transport services, the majority using the free, public funded '108'/'102' call centre-based ambulance services[19 23 24] and paid subsidised, commercial or personal transport. There are other services across states that, unlike '108', are basic patient transportation services with no en route stabilisation care. These may also provide transport between facilities.[21 25 26] For example, Janani Express yojana (public private non-ambulance transport service), operating through call centres within districts, transfers exclusively pregnant women and newborns in Madhya Pradesh and Odisha.[21 25] In larger cities, some other ambulance services (free or paid) are also available.

There have been very few research studies on transportation for IFTs of pregnant or postpartum women in India, despite the large investments made in public referral transport. This study was conducted to investigate IFTs of pregnant women in India using '108' ambulance services. The objectives of the research were to (i) estimate proportion of women transferred by '108' among all institutional deliveries in the general population, (ii) estimate the proportion of IFTs transferred by '108' among all institutional deliveries in the population, (iii) estimate the proportion of IFTs among all transfers of pregnant women by '108' and (iv) compare the characteristics of the IFTs and non-IFTs.

## METHODS
### Context
The '108' ambulance service operates under a public-private partnership. It operates 7361 ambulances, and transfers any medical emergency across 21 states and union territories (smaller less populated administrative units in India).[26] GVK-Emergency Management and Research Institute (GVK-EMRI) is the largest service provider for '108' and operates in 15 states and 2 union territories. In 2014–2015, GVK-EMRI '108' ambulances transferred about 3.6 million pregnant women to health facilities, which were about two-fifths of all the transfers by GVK-EMRI '108'.[23] There is approximately one ambulance for every 100 000 population, and the ambulance should be well equipped and accompanied by an Emergency Medicine Technician (EMT). The EMT is trained to provide emergency care and basic life support in obstetrics. In cases of imminent childbirth, the EMT is expected to assist the delivery en route and transfer the mother and child to the nearest health facility.[27] For IFTs, the EMT consults the '108' call centre-based medical officer and the referring healthcare provider to discuss indication of referral, stability of case to withstand travel, stabilising care required and place of referral (source: expert from GVK-EMRI). '108' service preferably transfers mother to public facilities; however, in absence of appropriate public facility nearby, they transfer the mother to empanelled private facility.[27]

### Working definitions
IFT, for this study, was defined as any transfer of a pregnant woman from one health facility to another health facility on the advice of a healthcare provider, using a '108' ambulance. All other transfers of pregnant women to health facilities using '108' ambulances were defined as 'non-IFT'. These mostly included transfers from home to facility.

An obstetric emergency was defined as any life-threatening medical complication in women in pregnancy, labour or childbirth, or after (within 42 days of termination of pregnancy). For IFT women, diagnosis of obstetric emergency was made by the referring doctor and noted by EMT after discussion with call centre-based doctor. For non-IFT, diagnosis was made by EMT. We observed that even high-risk such as previous caesarean section and precious pregnancy were considered obstetric emergency by '108' services thus we included them.[28]

### Study design
This cross-sectional study analysed '108' ambulance records from five states. Ethical approval for the study was obtained from Indian Institute of Public Health-Hyderabad and London School of Hygiene and Tropical Medicine.

### Study population
Pregnant women who called '108' or for whom a relative or friend or healthcare provider called on their behalf,

between 1 April 2013 and 31 March 2014 in the five states in India where GVK-EMRI operated '108' service had been fully functional for more than 3 years were included in this analysis (Andhra Pradesh, Chhattisgarh, Gujarat, Himachal Pradesh and Telangana). The period 1 April to 31 March was aligned with reporting period of state governments. States were chosen in a manner that they had representation of North, South, East and West of India.

## Obtaining data and data management

Formal permission to use the data was obtained from GVK-EMRI. Anonymised information on '108' calls from 1 April 2013 to 31 March 2014 was obtained from the GVK-EMRI emergency response centre database. Details on data management are described in another paper from the same dataset.[28] Data were extracted onto Excel sheets, and converted to STATA V.10.0 files. Data were inspected to assess consistency, range and missing data. Any gross issue relating to the quality of records was noted, and records with improbable entries were excluded from analysis. Variables were recoded wherever needed. Variables of interest were: IFT; age of pregnant woman; social caste (general, other backward, schedule tribe and scheduled caste); economic class (below or above poverty line); area (rural or urban); type of emergency; time of call; day of call; time taken by ambulance to reach the client; time taken to reach the health facility and distance travelled. Castes are classified based on historically backward and deprived social castes. The information was provided by the caller and EMT, and recorded in the call centre database. Districts within states were stratified into high priority districts and non-high priority districts. High priority districts are those identified by the Government of India as being in the lowest quartile of districts (or tribal districts) in each state with respect to maternal and child health indicators (including institutional delivery rate, maternal mortality and neonatal mortality rates among a total of 16 indicators).[29]

## Analysis

We used information from the Census 2011, District-Level Household Survey (DLHS) 2012–2013,[30–32] and Annual Health Survey (AHS) 2012–2013,[33] to estimate the number of pregnancies and institutional deliveries in the population for the study states, as mentioned below. The numbers of pregnant women transported by '108' as recorded in the call centre database were compared with these population estimates for each state.

For each state, the number of pregnancies in the study period was estimated as sum of estimated pregnancies in rural and urban population (population (rural)×crude birth rate (rural)×1.1×1000)+(population (urban)×crude birth rate (urban)×1.1×1000). The population data were obtained from the 2011 census and the crude birth rates from the Sample Registration System 2013. The multiplier 1.1 is used to account for an estimated 10% of the

pregnancies which may have ended in abortions or intra-uterine deaths.[34]

The number of institutional deliveries in the study period was estimated as (estimated number of pregnancies (rural)×institutional delivery rate (rural) ×100) + (estimated number of pregnancies (urban)×institutional delivery rate (urban)×100). Institutional delivery rates include live births and stillbirths. The institutional delivery rate were obtained from DLHS-4 and AHS-2 surveys.

Information was analysed for all states combined and separately by state, comparing IFT and non-IFT. The characteristics of the women transported, distances travelled and the time taken by '108' ambulances were described for both IFT and non-IFT journeys. The association between sociodemographic and clinical variables and the outcome (IFT vs no-IFT) was investigated using bivariate and multivariate logistic regression.

Social caste and economic class were missing in 55% and 95% of observations in the state of Chhattisgarh, and 14% and 22% in Himachal Pradesh. Given the nature of the variables, we decided that it was likely that the data were missing not at random, and therefore multiple imputation was considered inappropriate. In order to investigate possible selection bias resulting from missing data, we performed a complete case analysis for each state and all states combined, supplemented by a series of sensitivity analyses. Our first sensitivity analysis involved running a model with and without states of Chhattisgarh and Himachal Pradesh. Presence or absence of these states did not substantially change the pattern of results. The second sensitivity analysis was performed by running a model with and without social caste and economic class for all states combined (total) and for individual states. The presence or absence of these variables did not substantially change the magnitude of the pattern of results (ORs and $R^2$ for each model) for the total and for states (except Chhattisgarh). For our main analysis (all states combined), we thus included social caste and economic class variables and excluded any data from Chhattisgarh. Models for individual states (except Chhattisgarh) also include social caste and economic class.

## RESULTS
### Study populations

The study states had population sizes ranging from 6 to 60 million and had different social compositions (table 1). The percentage of rural population ranged from 61% in Telangana to 90% in Himachal Pradesh, while scheduled castes and tribes, together, ranged from 22% in Andhra Pradesh and Gujarat to 43% in Chhattisgarh. The crude birth rate was lowest in Himachal Pradesh (1.6%) and highest in Chhattisgarh (2.5%). The states also varied in the institutional delivery rates from 39.5% in Chhattisgarh to 94.1% in Telangana, and in their MMRs from 244 in Chhattisgarh to 92 per 1 00 000 live births in Telangana and Andhra Pradesh. On an average, '108' ambulance is

**Table 1** Demographic characteristics of the study states

| | Andhra Pradesh | Chhattisgarh | Gujarat | Himachal Pradesh | Telangana |
|---|---|---|---|---|---|
| Total population* | 49 386 799 | 25 540 196 | 60 383 628 | 6 856 509 | 35 193 978 |
| Rural/tribal | 70.4% | 76.8% | 57.4% | 90.0% | 61.3% |
| Urban | 29.6% | 23.2% | 42.6% | 10.0% | 38.7% |
| Scheduled caste* | 17.1% | 12.8% | 6.8% | 25.2% | 15.4% |
| Scheduled tribe* | 5.3% | 30.6% | 14.8% | 5.7% | 9.3% |
| Crude birth rate per 1000 population† | 17.5 | 24.5 | 21.1 | 16.2 | 17.5 |
| Institutional delivery rate per 100 childbirth‡ | 88.5% | 39.5% | 78.1% | 77.8% | 94.1% |
| MMR per 100 000 livebirths (2011–2013)† | 92 | 244 | 112 | Not available | 92 |
| Neonatal Mortality Rate per 1000 livebirths (2012–2013)† | 25 | 31 | 26 | 25 | 25 |
| No. of ambulances under '108'§ | 468 | 240 | 506 | 171 | 334 |
| Geographic region/terrain | Tribal pockets | Large tribal; hilly pockets | Tribal pockets | Majority hilly | Tribal pockets |

*Census 2011.

†Sample Registration System 2013—separate data for Telangana and Andhra Pradesh not available.

‡DLHS (2012–2013)/AHS (2011–2012).

§GVK-EMRI annual report for period April 2013–March 2014.

¶In Chhattisgarh, '102' ambulance service took over from October 2013 to March 2014.

AHS, Annual Health Survey; DLHS, District-Level Household Survey; GVK-EMRI, GVK-Emergency Management and Research Institute.

sanctioned for approximately 100 000 population in all states except Himachal Pradesh where one ambulance caters to about 40 000 population. Himachal Pradesh is mostly hilly and has sparse population distribution and '108' ambulances are stationed based on geographic regions.

### Description of 108 calls relating to pregnancy
Across the five states, '108' call centres received 6 46 656 calls for pregnancy-related transfers, 6.2% of which were for IFT. Among IFT calls, an ambulance was not assigned to 1.2%, and an ambulance was assigned but not used by 6.0% of callers. Among non-IFT callers, these proportions were 1.4% and 3.9%, respectively. A total of 6 08 559 pregnant women were transported using '108', and of these journeys 5.8% were for IFT.

### Estimated proportions of pregnancies, institutional deliveries and IFTs transported by '108'
Table 2 presents estimates of the number of pregnant women, obstetric emergencies, institutional deliveries and IFTs in the general populations, and the proportion of these transported using '108' ambulances in the five states, and overall. The 6 08 559 pregnant women transferred comprised 16.5% of all estimated pregnancies, and 20.8% of all estimated institutional deliveries for the study states combined.

Of the estimated institutional deliveries, '108' transported only 1.2% (34 993) women between facilities across states. This proportion was highest in Himachal Pradesh (3.0%) followed by Andhra Pradesh (1.7%), Telangana (1.4%), Chhattisgarh (0.7%) and lowest in Gujarat (0.6%). Only 1.0% (28 448) of all institutional deliveries were transported by '108' for obstetric emergencies—highest 2.4% in Himachal Pradesh and lowest 0.6% in Andhra Pradesh and Telangana (table 2).

### Characteristics of pregnant women transferred by '108' ambulances
The pregnant women who were transferred by '108' belonged mostly to lower social and economic sections—rural or tribal areas (84.6%) or scheduled castes or tribes (64.9%) and below-the-poverty-line status (76.6%). Two hundred and forty-two women (40 per 100 000) died before the ambulance reached the pick-up site. The proportion of pregnant women who died before arrival of '108' ambulance was higher in Chhattisgarh (150 per 100 000) compared with other states.

Of the pregnant women transferred by '108' ambulances, 34 993 (5.8%) had an IFT (table 3). The proportion of IFTs among women transported by '108' was highest in Himachal Pradesh (11.3%) followed by Andhra Pradesh (9.9%), Telangana (8.7%), Chhattisgarh (3.2%) and Gujarat (2.4%).

Overall, IFTs were made up of a higher proportion of younger women, women from backward caste, belonging to below-the-poverty-line and from urban areas compared with non-IFTs. A similar pattern was observed across all the states except in Himachal Pradesh where IFTs had lower proportion of women from below-the-poverty-line strata compared with non-IFTs (table 3). Delivery en route or in

**Table 2** Estimated proportion of pregnant women transported to hospitals by '108' from April 2013 to March 2014

| | Total | Andhra Pradesh | Chhattisgarh †† | Gujarat | Himachal Pradesh | Telangana |
|---|---|---|---|---|---|---|
| Estimated number of pregnancies in the population* | 3 811 920 | 950 696 | 678 217 | 1 387 021 | 121 638 | 673 508 |
| Estimated number of institutional deliveries in the population† | 2 921 674 | 842 109 | 267 896 | 1 083 264 | 94 634 | 633 771 |
| Proportion of all pregnancies transported by '108' ‡ | 16.5% | 15.5% | 9.0% | 19.5% | 20.5% | 15.7% |
| Proportion of all institutional deliveries transported by '108'§ | 20.8% | 17.5% | 22.7% | 24.9% | 26.3% | 16.6% |
| Proportion of all institutional deliveries that were transported by '108' for interfacility transfers¶ | 1.2% | 1.7% | 0.7% | 0.6% | 3.0% | 1.4% |
| Proportion of institutional deliveries that were transported by '108' for obstetric emergencies** | 1.0% | 0.6% | 1.0% | 1.4% | 2.4% | 0.6% |

*(Ppopulation (rural)×crude birth rate (rural)×1.1×1000)+(population (urban)×crude birth rate (urban)×1.1×1000); using census populations and crude birth rates from Sample Registration Survey in respective states.
†(Estimated number of pregnancies (rural)×institutional delivery rate (rural)×100)+(estimated number of pregnancies (urban)×institutional delivery rate (urban)×100); institutional delivery rates include live births and stillbirths. Using institutional delivery rates from District-Level Household Survey/Annual Household Survey in respective states.
‡Number of pregnant women transported by '108'/estimated number of pregnancies in the population.
§Number of pregnant women transported by '108'/estimated number of institutional deliveries in the population.
¶Number of pregnant women transported between facilities (IFTs) by '108'/estimated number of institutional deliveries in the population.
**Number of pregnant women transported by '108' for obstetric emergency/estimated number of institutional deliveries in the population.
††In Chhattisgarh, '102' ambulance service took over from '108' service from October 2013 to March 2014. Thus, number of beneficiaries of '108' and proportions may be underestimation of utilisation for annual estimates.

the ambulance was less likely among IFTs compared with non-IFTs across all states.

### Obstetric emergencies among women transferred by '108' ambulance for IFT and non-IFT

The majority of transfers by '108' ambulances were for normal labour. By state, between 2.7% and 9.3% of transferred pregnant women had an obstetric emergency; the overall average was 4.8%. The IFTs (8.4%) had higher proportion of pregnant women who had an obstetric emergency compared with non-IFTs (4.4%) (table 4). Himachal Pradesh had highest proportion of obstetric emergencies transferred for both IFT (21.0%) and non-IFT (7.8%). Andhra Pradesh was the lowest proportion of obstetric emergencies transferred for IFT (5.4%) and non-IFT (3.1%).

The most common obstetric emergencies were 'abnormal presentation of fetus' and 'bleeding in pregnancy' in Telangana, Andhra Pradesh and Gujarat; and 'bleeding in pregnancy' and 'medical conditions complicating pregnancy' in Himachal Pradesh and Chhattisgarh. IFTs had higher proportions of 'bleeding in pregnancy' cases across all the states and 'medical conditions complicating pregnancy' in Himachal Pradesh and Chhattisgarh (table 4). In Telangana and Andhra Pradesh, some IFTs (0.5% and 0.2%) and non-IFTs (8.43% and 0.53%) were for antenatal women requiring check-ups at higher institutions (data not shown).

### Destination facilities

About 86% of all transfers were to government or government-supported hospitals. This figure was highest in Himachal Pradesh (97%) and lowest in Andhra Pradesh, where around 82% of women were transferred to government hospitals (table 4). With type of government facility, across the states, half of the IFTs were to district-level secondary or tertiary hospitals, while non-IFTs were to subdistrict area or civil hospitals or Community Health Centres. Gujarat reported highest percentage of transfers to middle-level facilities. Himachal Pradesh had more transfers to civil hospitals or higher (table 4).

### Distance and time travelled by 108 ambulance for pregnant women

The median distances travelled and time taken by '108' ambulances to transfer pregnant women are shown in table 4. Ambulances travelled less than 4 km to reach half of the pregnant women for IFTs across four states but up to 10 km in Andhra Pradesh. For non-IFTs, ambulances travelled farther (between 9 and 12 km) to reach half of the pregnant women. Among IFTs, median distances from pick-up site to destination facility were between 16 km in Chhattisgarh to 32 km in Himachal Pradesh, but nearer in non-IFTs (between 8 and 32 km).

The median time taken for the ambulance to travel to the pick-up site for IFTs was between 10 and 15 min

**Table 3** Demographic characteristics of users of '108' ambulance service for IFT and non-IFT in states (April 2013 to March 2014)

| | Total | | Andhra Pradesh | | Chhattisgarh | | Gujarat | | Himachal Pradesh | | Telangana | |
|---|---|---|---|---|---|---|---|---|---|---|---|---|
| | IFT (n=34993) (5.8%) | Non-IFT (n=573566) (94.2%) | IFT (n=14574) (9.9%) | Non-IFT (n=132800) (90.1%) | IFT (n=1929) (3.2%) | Non-IFT (n=58881) (96.8%) | IFT (n=6562) (2.4%) | Non-IFT (n=263509) (97.6%]) | IFT (n=2809) (11.3%) | Non-IFT (n=22114) (88.7%) | IFT (n=9119) (8.7%) | Non-IFT (n=96262) (91.3%) |
| **Age group (years)** | | | | | | | | | | | | |
| ≤19 | 4.1 | 3.4 | 5.0 | 4.8 | 7.1 | 5.8 | 2.3 | 2.0 | 4.4 | 5.7 | 3.5 | 3.6 |
| 20–24 | 58.7 | 56.5 | 64.9 | 65.7 | 54.9 | 56.4 | 52.0 | 51.9 | 48.3 | 51.6 | 58.6 | 58.9 |
| 25–30 | 28.1 | 30.6 | 24.0 | 24.9 | 25.0 | 27.9 | 30.1 | 33.2 | 34.5 | 32.0 | 31.4 | 32.2 |
| 30–34 | 6.5 | 7.2 | 4.2 | 3.3 | 8.9 | 7.2 | 11.3 | 9.9 | 9.4 | 8.0 | 4.8 | 4.1 |
| ≥35 | 2.6 | 2.3 | 1.9 | 1.2 | 4.1 | 2.7 | 4.3 | 3.0 | 3.4 | 2.7 | 1.7 | 1.2 |
| Total non-missing | (32 538) 100 | (551 568) 100 | (13 105) 100 | (120 155) 100 | (1914) 100 | (58 591) 100 | (6551) 100 | (263 508) 100 | (2779) 100 | (21 694) 100 | (8178) 100 | (87 620) 100 |
| Missing | (2455) 7.0 | (21 988) 3.8 | (1469) 10.1 | (12 645) 9.5 | (15) 0.8 | (290) 0.5 | (0) 0 | (1) 0 | (30) 1.1 | (420) 1.9 | (941) 10.3 | (8642) 9.0 |
| **Social caste** | | | | | | | | | | | | |
| General caste | 12.2 | 11.3 | 10.8 | 10.9 | 3.7 | 3.5 | 14.8 | 12.8 | 37.3 | 38.2 | 7.1 | 5.0 |
| Other backward | 40.0 | 38.5 | 41.7 | 44.1 | 48.5 | 43.8 | 38.3 | 35.0 | 17.4 | 11.5 | 43.4 | 43.5 |
| Scheduled caste | 29.1 | 22.1 | 34.7 | 34.1 | 14.3 | 14.7 | 11.2 | 11.1 | 37.2 | 42.8 | 32.2 | 34.2 |
| Scheduled tribe | 18.7 | 28.2 | 12.8 | 10.9 | 33.6 | 38.1 | 35.8 | 41.2 | 8.1 | 7.6 | 17.3 | 17.3 |
| Total non-missing | (32 800) 100 | (535 328) 100 | (14 400) 100 | (131 924) 100 | (652) 100 | (27 427) 100 | (6562) 100 | (263 204) 100 | (2201) 100 | (17 148) 100 | (8996) 100 | (95 625) 100 |
| DK/missing | (2193) 6.3 | (38 238) 6.7 | (174) 1.2 | (876) 0.7 | (1277) 66.2 | (31 454) 53.4 | (11) 0.2 | (305) 0.1 | (608) 21.6 | (4966) 22.5 | (123) 1.3 | (637) 0.7 |
| **Economic class** | | | | | | | | | | | | |
| BPL | 87.7 | 75.9 | 99.7 | 99.6 | 96.7 | 97.0 | 54.2 | 55.9 | 62.7 | 67.0 | 99.5 | 99.5 |
| Others | 12.3 | 24.1 | 0.3 | 0.4 | 3.3 | 3.0 | 45.7 | 44.1 | 37.3 | 33.0 | 0.5 | 0.5 |
| Total non-missing | (32 296) 100 | (511 447) 100 | (14 357) 100 | (131 391) 100 | (61) 100 | (2824) 100 | (6540) 100 | (262 991) 100 | (2395) 100 | (19 041) 100 | (8943) 100 | (95 200) 100 |
| DK/missing | (2697) 7.7 | (62 119) 10.8 | (217) 1.5 | (1409) 1.1 | (1868) 96.8 | (56 057) 95.2 | (22) 0.3 | (518) 0.2 | (414) 14.7 | (3073) 13.9 | (176) 1.9 | (1062) 1.1 |
| **Area** | | | | | | | | | | | | |
| Rural/tribal | 68.7 | 85.6 | 67.4 | 79.5 | 92.1 | 92.7 | 62.8 | 89.3 | 75.9 | 95.2 | 67.6 | 77.8 |
| Urban | 31.3 | 14.4 | 32.6 | 20.5 | 7.9 | 7.3 | 37.2 | 10.7 | 24.1 | 4.8 | 32.4 | 22.2 |
| Total non-missing | (34 910) 100 | (570 042) 100 | (14 555) 100 | (132 510) 100 | (1929) 100 | (58 881) 100 | (6562) 100 | (263 509) 100 | (2770) 100 | (18 952) 100 | (9094) 100 | (96 190) 100 |
| Missing | (78) 0.2 | (3524) 0.6 | (19) 0.1 | (290) 0.2 | (0) 0 | (0) 0 | (0) 0 | (0) 0 | (39) 1.4 | (3162) 14.3 | (25) 0.3 | (72) 0.1 |
| Died before ambulance reached | 0.07 | 0.04 | 0.04 | 0.02 | 0.36 | 0.14 | 0.02 | 0.03 | 0 | 0.01 | 0.02 | 0.02 |
| **Delivery en route** | | | | | | | | | | | | |
| At pick-up site | 0.3 | 1.4 | 0.2 | 0.8 | 1.2 | 2.8 | 0.5 | 1.7 | 0.2 | 1.7 | 0.2 | 0.7 |
| In ambulance | 0.6 | 1.0 | 0.4 | 0.7 | 0.8 | 1.5 | 0.6 | 0.9 | 1.7 | 3.5 | 0.4 | 0.7 |

BPL, below the poverty line; IFT, interfacility transfer.

**Table 4** Characteristics of transfers for IFT and non-IFT by state (April 2013 to March 2014)

| | Total | | Andhra Pradesh | | Chhattisgarh | | Gujarat | | Himachal Pradesh | | Telangana | |
|---|---|---|---|---|---|---|---|---|---|---|---|---|
| | IFT (n=34 993) | Non-IFT (n=573 566) | IFT (n=14 574) | Non-IFT (n=132 800) | IFT (n=1929) | Non-IFT (n=58 881) | IFT (n=6562) | Non-IFT (n=263 509) | IFT (n=2809) | Non-IFT (n=22 114) | IFT (n=9119) | Non-IFT (n=96 262) |
| Obstetric emergency, % | 8.4 | 4.4 | 5.4 | 3.1 | 18.0 | 3.9 | 10.4 | 5.5 | 21.0 | 7.8 | 5.8 | 3.2 |
| Abnormal presentation | 1.5 | 2.0 | 1.0 | 1.0 | 1.3 | 1.2 | 3.2 | 3.1 | 0.7 | 0.4 | 1.3 | 1.1 |
| Bleeding in pregnancy | 3.5 | 0.9 | 2.0 | 0.6 | 1.6 | 0.4 | 4.5 | 1.0 | 14.1 | 5.6 | 2.1 | 0.6 |
| Eclampsia/convulsion | 0.7 | 0.1 | 0.6 | 0.2 | 0.7 | 0.1 | 0.6 | 0.1 | 1.1 | 0.1 | 0.8 | 0.2 |
| Fetal loss | 0.2 | 0.1 | 0.1 | 0.1 | 0.6 | 0.4 | 0.1 | 0.02 | 0.4 | 0.1 | 0.1 | 0.1 |
| Medical condition | 1.8 | 0.7 | 0.9 | 0.6 | 13.6 | 1.7 | 1.0 | 0.4 | 3.8 | 1.1 | 0.8 | 0.8 |
| Previous caesarean | 0.4 | 0.4 | 0.5 | 0.3 | 0.3 | 0.2 | 0.5 | 0.5 | 0.3 | 0.3 | 0.4 | 0.3 |
| Precious pregnancy | 0.3 | 0.3 | 0.2 | 0.2 | 0.1 | 0.1 | 0.5 | 0.4 | 0.5 | 0.2 | 0.3 | 0.2 |
| Other emergency, % | 3.7 | 1.8 | 4.1 | 2.8 | 8.7 | 1.2 | 2.3 | 1.0 | 0 | 0 | 4.2 | 3.5 |
| Destination facility, % | | | | | | | | | | | | |
| Government | 82.0 | 79.6 | 82.1 | 81.7 | 87.6 | 96.8 | 70.2 | 72.0 | 96.0 | 97.6 | 85.3 | 84.1 |
| District secondary or tertiary hospital | 53.2 | 19.8 | 54.2 | 29.4 | 64.3 | 28.5 | 30.1 | 7.3 | 86.9 | 46.7 | 55.7 | 32.0 |
| Area/civil hospital | 19.8 | 13.4 | 18.1 | 21.1 | 0.6 | 0.8 | 31.5 | 8.3 | 7.7 | 27.3 | 21.3 | 22.5 |
| Community Health Centre | 7.6 | 34.6 | 8.3 | 19.7 | 20.5 | 57.4 | 7.4 | 43.6 | 0.9 | 20.4 | 6.3 | 16.2 |
| Primary Health Centre | 1.1 | 9.0 | 1.2 | 11.3 | 1.9 | 9.3 | 0.2 | 7.0 | 0.3 | 3.2 | 1.9 | 13.1 |
| Sub-Health Centre | 0 | 0.1 | 0 | 0 | 0.3 | 0.9 | 0 | 0 | 0 | 0 | 0 | 0 |
| Information missing | 0.4 | 2.8 | 0.3 | 0 | 0 | 0 | 0.9 | 5.6 | 0 | 0 | 0.2 | 0.2 |
| Government supported | 3.4 | 7.7 | 0 | 0 | 0 | 0 | 16.9 | 15.9 | 0 | 0 | 0 | 0 |
| Private | 14.5 | 12.8 | 17.9 | 18.3 | 12.4 | 3.2 | 12.9 | 12.1 | 4.0 | 2.4 | 14.7 | 15.9 |
| Total non-missing | (31 911) 100 | (531 392) 100 | (12 693) 12.9 | (116 028) 100 | (1784) 100 | (54 093) 100 | (6442) 100 | (256 898) 100 | (2777) 100 | (21 449) 100 | (8215) 100 | (82 924) 100 |
| Missing | (3082) 8.8 | (42 174) 7.4 | (1881) 12.9 | (16 772) 12.6 | (145) 7.5 | (4788) 8.1 | (120) 1.8 | (6611) 2.5 | (32) 1.1 | (665) 3.0 | (904) 9.9 | (13 338) 13.9 |
| To different district | 32.1 | 28.5 | 29.8 | 28.8 | 62.5 | 54.3 | 17.7 | 22.0 | 31.6 | 10.1 | 39.7 | 34.3 |
| To different mandal | 86.8 | 55.9 | 89.5 | 75.7 | 83.2 | 68.5 | 74.1 | 33.0 | 96.3 | 93.5 | 89.5 | 74.9 |
| Distance call to pick-up site* (km) Median (IQR) | 2 (1–14) | 12 (6–18) | 10 (3–18) | 12 (6–19) | 4 (1–6) | 9 (2–17) | 1 (1–10) | 12 (7–18) | 1 (1–1) | 11 (6–18) | 3 (1–15) | 12 (6–20) |
| Distance pick-up site to hospital* (km) Median (IQR) | 28 (18–40) | 15 (8–23) | 21 (12–31) | 17 (10–26) | 16 (5–30) | 8 (3–15) | 24 (14–37) | 14 (8–21) | 32 (21–48) | 12 (6–20) | 29 (19–43) | 17 (10–28) |
| Time call to pick-up site* (min) Median (IQR) | 11 (7–28) | 25 (16–37) | 24 (12–37) | 27 (17–40) | 15 (7–33) | 27 (17–39) | 10 (7–19) | 24 (16–34) | 12 (8–27) | 35 (22–56) | 12 (7–29) | 26 (16–39) |
| Time pick-up site to hosp* (min) Median (IQR) | 49 (32–70) | 26 (16–40) | 37 (23–55) | 32 (20–46) | 40 (26–64) | 21 (13–32) | 40 (26–60) | 23 (14–34) | 83 (57–116) | 43 (28–65) | 48 (33–70) | 31 (20–48) |
| Time call to hospital* (min) Median (IQR) | 77 (56–102) | 63 (46–84) | 75 (55–97) | 72 (54–93) | 70 (50–97) | 56 (39–77) | 64 (47–86) | 56 (42–74) | 105 (72–143) | 80 (54–121) | 77 (57–103) | 70 (52–93) |

*N varies, excludes deliveries by EMT that were not transported or missing value.
EMT, Emergency Medicine Technician; IFT, interfacility transfer.

for four states but longer (24 min) in Andhra Pradesh. The median time taken for travel from the referring to the destination facility was <40 min in Andhra Pradesh, Chhattisgarh and Gujarat but longer for Telangana (48 min) and Himachal Pradesh (83 min). For non-IFT, these times were about 30 min each, except for Himachal Pradesh where travel times were longer (table 4).

### Determinants of IFT

Overall, women with obstetric emergencies transported by 108 were roughly twice as likely to have an IFT as women with no obstetric emergency (crude OR 2.25, 95% CI 2.16 to 2.34) (table 5). In the adjusted analysis (excluding Chhattisgarh), obstetric emergencies had 1.95 (95% CI 1.83 to 2.06) times higher odds of having IFT compared with non-emergencies (table 5). Women from urban areas were twice as likely to have IFTs as women in rural areas (adjusted OR (AOR) 2.34, 95% CI 2.26 to 2.40). There was no evidence of an independent effect of high-priority districts on IFT. Overall, there was evidence of a 'J'-shaped trend with maternal age. IFTs were marginally more likely in women <25, with a trend of increasing odds in women aged 30 yeas or more, compared with women aged 25–30 years (AOR 1.01, 1.04, 1, 1.21, 1.43, respectively for age groups <19, 20–24, 25–30, 30–34 and ≥35 years). Although the effects were small, women from backward castes (AOR 0.95, 95% C.I. 0.92 to 0.99), scheduled castes (AOR 0.94, 95% C.I. 0.90 to 0.98) and scheduled tribes (AOR 0.98, 95% C.I. 0.95 to 1.00) were less likely to have IFT than women from general castes, after adjustment. Women from below the poverty line (AOR 0.92, 0.88–0.96) were less likely to have IFTs compared with women above the poverty line.

The determinants of IFT had different pattern of effect across states as shown in table 5. In respect to individual state models, the association between obstetric emergencies and IFT was strongest for Chhattisgarh (AOR 5.32, 95% CI 4.70 to 6.02) followed by Himachal Pradesh (AOR 3.03, 95% CI 2.65 to 3.47).

### DISCUSSION

This is the first study assessing IFTs for pregnant women using the '108' ambulance service—the largest provider of the emergency medical services in India. We discuss findings with respect to patterns of use and the existing health system.

We estimated that '108' transferred around one-fifth of all pregnancies and institutional deliveries in the five states. However, '108' service transported only 1% of all institutional deliveries in the population for IFTs. Only 1% of all institutional deliveries were transported by '108' for obstetric emergencies. The findings suggest that the '108' service is not a preferred choice for transport from lower-level facility to a higher-level facility, or for obstetric emergencies. Details about characteristics of obstetric emergencies transported by '108' ambulance services

and discussion about coverage are published in another paper.[28]

The proportion of IFTs among all institutional deliveries will depend on the pattern of use of level of healthcare, referral practices and the availability of transport for between facility transfers. Roughly one-half of the non-IFTs went to peripheral birthing centres or basic Emergency Obstetric Care (EmOC) centres in our study. A systematic review from India (including most studies from public health facilities) found that between 14% and 36% of all pregnancies were referred from nurse-run delivery or basic EmOC centres, and a further 2%–7% were referred from doctor-run basic EmOC centres for complications or emergencies.[13] Assuming the pattern of use of health facilities in our study and evidence from the review, we estimate that between 40 000 and 80 000 institutional deliveries who used '108' for non-IFTs may require further referral to higher facility. In addition, among the estimated 80% (2 300 000) institutional deliveries who went to their first facility by other means of transport, some women may be referred further. Thus, the absolute numbers of pregnant women referred and requiring transport for IFT are likely to be large while '108' transports only about 35 000 pregnant women for IFT.

In our analysis, of all the transfers by '108' only 5.8% were IFTs—lowest 2.4% in Gujarat and highest 11.3% in Himachal Pradesh. The proportion of IFT was higher in states of Karnataka and Tamil Nadu (12.8% and 35.7%) among '108' users in 2013–2014.[24] It appears that there is potential for increasing use of '108' for IFTs in the study states.

A study conducted in Andhra Pradesh in 2009 found that none of the pregnant women who used '108' was referred from a facility.[35] Some women did not prefer to wait for '108' if they perceived any emergency.[35] A maternal death review in Uttar Pradesh in 2010 found that only 5 of the 32 mothers who were transferred between facilities used an ambulance.[6] However, for other public transportation schemes, it was found that a high proportion (two-thirds) of all the interfacility referrals in a study from Madhya Pradesh used Janani Express service (non-ambulance) while others used personal transport, taxis, autorikshaws or public transport.[10]

Although IFTs in our study were twice as likely to transport pregnant women who had any obstetric emergency compared with non-IFTs, there was a very large proportion of IFTs with no obstetric emergency or complication (92%). One of the '108' doctors, during discussion to understand IFT processes, mentioned that on many occasions they were not convinced of the need for IFT. However on insistence of the referring staff, the '108' doctor approved transport for IFT (personal communication). Often the referral was done because there was no doctor on duty or other resources were not available, as was also found in Madhya Pradesh.[10] These non-emergency IFTs will add unnecessary load at higher facilities and also make the ambulances unavailable for other emergencies.

**Table 5** Determinants of IFT across states (logistic regression analysis showing OR (95% CI))

| | Total* | Andhra Pradesh | Chhattisgarh | Gujarat | Himachal Pradesh | Telangana |
|---|---|---|---|---|---|---|
| **Bivariate logistic regression: OR (95% CI)** | | | | | | |
| Obstetric emergency | 2.25 (2.16 to 2.34) | 1.81 (1.67 to 1.95) | 5.39 (4.76 to 6.09) | 2.01 (1.85 to 2.17) | 3.15 (2.84 to 3.49) | 1.90 (1.73 to 2.09) |
| Urban area | 2.29 (2.23 to 2.34) | 1.87 (1.80 to 1.94) | 1.09 (0.92 to 1.29) | 4.93 (4.69 to 5.20) | 6.34 (5.68 to 7.07) | 1.68 (1.60 to 1.76) |
| High priority district | 0.96 (0.94 to 0.98) | 1.10 (1.06 to 1.14) | 1.28 (1.13 to 1.44) | 0.69 (0.65 to 0.73) | 0.92 (0.83 to 1.01) | 0.95 (0.90 to 1.00) |
| **Age group (years)** | | | | | | |
| ≤19 | 1.06 (1.00 to 1.13) | 1.08 (0.98 to 1.18) | 1.37 (1.13 to 1.66) | 1.28 (1.08 to 1.51) | 0.70 (0.57 to 0.86) | 1.01 (0.88 to 1.14) |
| 20–24 | 1.03 (1.00 to 1.05) | 1.02 (0.98 to 1.07) | 1.09 (0.97 to 1.21) | 1.10 (1.04 to 1.16) | 0.87 (0.79 to 0.95) | 1.02 (0.97 to 1.07) |
| 25–30 | 1 | 1 | 1 | 1 | 1 | 1 |
| 30–34 | 1.24 (1.18 to 1.30) | 1.29 (1.17 to 1.42) | 1.38 (1.16 to 1.65) | 1.25 (1.15 to 1.36) | 1.08 (0.93 to 1.25) | 1.21 (1.08 to 1.35) |
| >35 | 1.52 (1.41 to 1.64) | 1.62 (1.41 to 1.86) | 1.68 (1.31 to 2.15) | 1.58 (1.39 to 1.80) | 1.16 (0.93 to 1.46) | 1.47 (1.23 to 1.76) |
| **Social caste** | | | | | | |
| General caste | 1 | 1 | 1 | 1 | 1 | 1 |
| Backward caste | 0.92 (0.89 to 0.96) | 0.95 (0.90 to 1.01) | 1.04 (0.68 to 1.58) | 0.94 (0.87 to 1.02) | 1.55 (1.36 to 1.77) | 0.71 (0.65 to 0.77) |
| Scheduled caste | 0.91 (0.87 to 0.94) | 1.02 (0.97 to 1.09) | 0.91 (0.58 to 1.44) | 0.87 (0.79 to 0.96) | 0.89 (0.80 to 0.99) | 0.67 (0.61 to 0.73) |
| Scheduled tribe | 0.90 (0.87 to 0.94) | 1.18 (1.11 to 1.28) | 0.83 (0.54 to 1.27) | 0.75 (0.69 to 0.81) | 1.10 (0.93 to 1.31) | 0.71 (0.65 to 0.78) |
| Below poverty line | 0.92 (0.88 to 0.96) | 1.45 (1.04 to 2.01) | 0.92 (0.22 to 3.80) | 0.93 (0.89 to 0.98) | 0.83 (0.76 to 0.90) | 1.03 (0.75 to 1.41) |
| **Multivariate logistic regression model: AOR (95% CI)†** | | | | | | |
| Obstetric emergency† | 1.96 (1.83 to 2.06) | 1.80 (1.65 to 1.95) | 5.32 (4.70 to 6.02) | 1.89 (1.74 to 2.05) | 3.03 (2.65 to 3.47) | 1.79 (1.61 to 1.98) |
| Urban area | 2.35 (2.26 to 2.41) | 1.86 (1.79 to 1.94) | 1.09 (0.92 to 1.29) | 4.75 (4.50 to 5.01) | 5.76 (5.03 to 6.59) | 1.64 (1.56 to 1.73) |
| High priority district | 0.98 (0.95 to 1.00) | 1.09 (1.05 to 1.13) | 1.25 (1.11 to 1.41) | 0.90 (0.85 to 0.96) | 1.17 (1.03 to 1.33) | 1.00 (0.95 to 1.06) |
| **Age group (years)** | | | | | | |
| ≤19 | 1.01 (0.94 to 1.06) | 1.01 (0.92 to 1.11) | 1.36 (1.12 to 1.65) | 1.25 (1.06 to 1.49) | 0.82 (0.65 to 1.05) | 0.91 (0.80 to 1.05) |
| 20–24 | 1.04 (0.98 to 1.06) | 1.03 (0.99 to 1.08) | 1.11 (1.00 to 1.24) | 1.12 (1.06 to 1.18) | 0.92 (0.82 to 1.03) | 1.02 (0.97 to 1.07) |
| 25–30 | 1 | 1 | 1 | 1 | 1 | 1 |
| 30–34 | 1.21 (1.14 to 1.27) | 1.26 (1.14 to 1.39) | 1.33 (1.11 to 1.59) | 1.21 (1.11 to 1.32) | 0.94 (0.77 to 1.14) | 1.19 (1.07 to 1.34) |
| >35 | 1.43 (1.35 to 1.56) | 1.56 (1.36 to 1.80) | 1.52 (1.19 to 1.95) | 1.44 (1.27 to 1.64) | 0.96 (0.72 to 1.30) | 1.43 (1.19 to 1.71) |
| **Social caste** | | | | | | |
| General caste | 1 | 1 | NA‡ | 1 | 1 | 1 |
| Backward caste | 0.95 (0.92 to 0.99) | 0.93 (0.87 to 0.99) | NA‡ | 1.02 (0.94 to 1.10) | 1.80 (1.55 to 2.10) | 0.79 (0.71 to 0.87) |
| Scheduled caste | 0.94 (0.90 to 0.98) | 1.02 (0.95 to 1.08) | NA‡ | 0.99 (0.90 to 1.09) | 1.01 (0.90 to 1.14) | 0.74 (0.61 to 0.73) |
| Scheduled tribe | 0.98 (0.96 to 1.00) | 1.22 (1.13 to 1.31) | NA‡ | 0.94 (0.87 to 1.01) | 1.31 (1.08 to 1.61) | 0.82 (0.67 to 0.82) |
| Below poverty line | 0.92 (0.88 to 0.96) | 1.40 (1.00 to 1.96) | NA‡ | 0.95 (0.90 to 1.00) | 0.85 (0.77 to 0.95) | 1.34 (0.95 to 1.88) |
| Pseudo-R² | 0.076 | 0.014 | 0.033 | 0.054 | 0.085 | 0.010 |

*Adjusted for state and does not include the state of Chhattisgarh.
†Adjusted for obstetric emergency, urban area, high priority district and age group, as appropriate.
‡For Chhattisgarh, social caste and below poverty line were not included.
AOR, adjusted OR; IFA, interfacility transfer; NA, not available.

In our study, IFTs had a higher overall proportion of women from disadvantaged social castes, below-the-poverty-line and urban areas compared with non-IFTs. Greater proportions of urban women deliver in health facilities,[30–32] and thus have a higher probability of early detection of complications and referral to a higher facility. Studies have also shown that higher proportions of women who use publically financed transportation schemes ('108'/'102'/Janani Express yojana) belong to historically disadvantaged and backward social caste, below the poverty line, are less educated and mostly from rural geographical areas compared with non-users.[21 35–38]

The median time from call to '108' and reaching the destination facility was similar for non-IFT, and IFT, and it ranged from 1 to 1.3 hour. A study of 57 maternal death reviews from Uttar Pradesh found much longer times; the mean time taken to arrange transport and travel from home to facility was about 4 hours, and transport from one facility to another was about 10 hours.[6] Although the '108' service has a mandate to inform the destination facility before arrival to reduce delays in treatment, this is not practiced as the list of contact points is not provided to EMTs (source: Dr GVR Rao). Few women were transferred to a district different from the originating district due to sparse distribution of referral facilities. These women also travelled longer distances. Continuity of care and monitoring of IFTs can be better if transfers are in same administrative unit that is, district.

The '108' ambulances are stationed close to CHCs and should be readily available, but if the ambulance is on route for pick-up or drop-off of another client then the IFT client with complications will have to wait or arrange for another means of transportation.[21] The '108' service may consider prioritising IFTs by having dedicated fleet of ambulances for IFTs over transporting all non-IFTs by increasing the fleet of ambulances stationed closer to beneficiaries.[25] In some states, the underused ambulances at the health facilities are being used through the '102' call centre along with '108'; however, the distribution of these into IFT and non-IFT use has not been well laid out.[21] In other states, '102' has new ambulances that focus on IFT or transfers back to home for pregnant women.[26] Since 2013, state of Assam has a dedicated fleet of 450 ambulances for IFTs operating through the '102' call system. Of these, over half of all IFTs were for pregnant women in 2015–2016 (source: GVK-EMRI annual statistics). This stresses the need for dedicated ambulances to deal with emergency IFTs, equipped with advanced life and obstetric support facilities. Other services like Janani Express yojana, do not provide clinical support during transfer and are found to take longer times in transfer of pregnant women.[21 25]

The role of the '108' service in improving care at the referring and referral facility, and its overall impact on maternal morbidity and mortality reduction, could not be estimated from our analysis nor has been reported in other studies. Nonetheless, the '108' service is accepted as an important and successful intervention to improve patient transportation for obstetric, medical and other emergencies.[21 24 26 35 37] A study from five states, showed that '108' ambulances provide prehospital stabilising care to all pregnant women, and delivery and postdelivery care to those delivered at home or on the ambulance.[39] However, a study from Punjab found that about half of '108' ambulances did not meet required standards for basic life support.[37] Monitoring of the quality of en route care and referral systems is integral for better outcomes and monitoring of any transport intervention.[16 21 40]

As we included representative states from all regions of India and included universal sample, our results are generalisable to the all the pregnant women transported by '108' ambulance service in India. There are a few limitations to our analysis. Details of the type of emergency in the '108' database was based on the doctor's report or the diagnoses by the EMTs for IFTs, and only the EMTs for non-IFTs. This information is thus subject to interobserver variability and differential reporting. However, we cannot estimate if this would have led to overestimation or underestimation of the effect estimate. A very few women who used '108' for antenatal care may have used it again for delivery care, and would have been counted twice. The database did not have information if the women was referred after childbirth thus postpartum referrals could not be computed separately. Data for treatment given en route, and doctors' notes on IFTs, were mostly not recorded in the '108' database and thus could not be studied. Details of the source hospital for IFTs were not available and thus details of transfers between type of facilities could not be assessed. There is a possibility that few calls for IFT for the women residing in rural areas were wrongly recorded, as they called from the health facilities in urban areas. This would have contributed to higher proportion of urban women among IFTs. The '108' service from Chhattisgarh was taken over by the '102' service since October 2013 thus use of '108' may be underestimated for this state. We had large proportions of missing information on social and economic status from two states. We considered that the missingness was not at random and was not associated with outcome. There is evidence that in such situations a complete case analysis, as we reported, is associated with negligible bias compared with a multiple imputation approach.[41]

## CONCLUSION

Of all the estimated institutional deliveries in India, only a very small proportion (6%) made use of the '108' ambulance for transfer between facilities. Among '108' users for IFTs, around 92% did not have any complication or emergency. After adjusting for confounding factors, IFTs were more likely for women with obstetric emergencies, more than 30 years of age and from urban areas. Pregnant women from socially disadvantaged castes, below poverty line and from high priority districts were less likely to have IFTs. Utilisation of the '108' service and its determinants varied across states. Primary research

is required to understand variation in utilisation and to explore the potential of the '108' ambulance service to serve a higher proportion of women requiring IFTs, in particular those having obstetric emergencies. The '108' service would benefit by having a triage system to ensure that women with an obstetric emergency requiring an IFT are prioritised.

**Author affiliations**
[1]Indian Institute of Public Health-Hyderabad, Public Health Foundation of India, Hyderabad, India
[2]Department of Non-Ccommunicable Disease Epidemiology, Faculty of Epidemiology and Population Health, London School of Hygiene and Tropical Medicine, London, UK
[3]Department of Infectious Disease Epidemiology, Faculty of Epidemiology and Population Health, London School of Hygiene and Tropical Medicine, London, UK
[4]Emergency Medicine Learning Center and Research, GVK-Emergency Management and Research Institute, Hyderabad, India
[5]Department of Clinical Research, Faculty of Infectious and Tropical Diseases, London School of Hygiene and Tropical Medicine, London, UK

**Acknowledgements** The authors acknowledge GVK-Emergency Management and Research Institute for sharing the data and allowing scientific study without any interference.

**Contributors** Conceived and designed the protocol: SS, PD, OC, GVR, GVSM. Contributed in analysis plan: SS, PD, OC, LO, GVSM. Analysed and interpreted the data: SS. Contributed in preparation of manuscript: SS, PD, OC, LO, GVR, GVSM.

**Funding** This work was supported by a Wellcome Trust Capacity Strengthening Strategic Award to the Public Health Foundation of India and a consortium of the UK universities.

**Competing interests** None declared.

**Patient consent** The study involved analysis of data saved in a call centre database. Personal identifiers were removed before analysis and reporting.

**Ethics approval** The research obtained ethics approval from ethics committees of both LSHTM and IIPH-Hyderabad (LSHTM Ethics Ref: 9613; IIPHH Ethics Ref: IIPHH/TRC/IEC/009/2014).

**Provenance and peer review** Not commissioned; externally peer reviewed.

**Data sharing statement** The data were obtained and analysed within the premises of GVK-EMRI office under license for the current study. The data are not publically available; however, can be available on reasonable request directly to GVK-EMRI.

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
