## [Reviewer comments · BMJ Open]

ARTICLE DETAILS

TITLE (PROVISIONAL)	Inter-facility transfer of pregnant women using publicly-funded emergency call-centre based ambulance services: A cross-sectional analysis of service logs from five states in India
AUTHORS	Singh, Samiksha; Doyle, Pat; Campbell, Oona; Oakley, Laura; Rao, G.V. Ramana; Murthy, GVS

VERSION 1 - REVIEW

REVIEWER	Lee Wallis University of Cape Town south africa
REVIEW RETURNED	15-Nov-2016

GENERAL COMMENTS	an interesting and well written paper covering an important topic in MCH IFT is spelled out multiple times before being abbreviated my main issues are twofold the first is length: i am concerned it is very long for what is basically a report of EMS case types (4800+ words) It could easily be cut down significantly the second lies in the methods: there is no justification for the dates chosen, and little detail about how and why the 5 states were chosen (other than that they had EMRI for a seemingly arbitrary 3 years). does this mean that all other states had EMRI for less time? does that matter? why 3 years? page 11 line 21-23: how valid is this 10% multiplier assumption? line 35-37: how valid is this assumption? otherwise i am happy with the paper
---

REVIEWER	Wally Carlo University of Alabama at Birmingham/ United States of America
REVIEW RETURNED	23-Nov-2016

GENERAL COMMENTS	This is an important manuscript as it reports a cross sectional analysis of a large emergency maternal ambulance service in large populations. Title "108" is not a universally-used concept. Maybe a descriptor that is of more general use such as "emergency free ambulances" or "emergency call center ambulances" could be used instead in the
---

	title and text. Abstract Include the % rather than just interpret as “very small” the proportion that used the ‘108’ ambulance for transfer between facilities. This is also pertinent to the last paragraph of the manuscript. Background The Background section includes details that are not directly related to the research presented and that are less generalizable to other settings such as alternative transport systems in various Indian states. Thus, the 4th paragraph should be shortened substantially. Methods Whenever possible, generic names and not commercial names, such as GVK-EMRI, should be used. It would be good to have some explanation of the classification of castes for an international journal. Results Table 1 should include units for all variables (including MMR and NMR) not just for some. Data on the time of days of the transports is commented on in the Discussion but should be presented in Results. Conclusions Minor Many abbreviations are not spelled out.
--	--

REVIEWER	Jennifer A. Newberry Stanford School of Medicine I regularly conduct research with GVK EMRI and one of the authors (Dr. GV Ramana Rao), but I was not involved in this study or work in anyway. I do not consider this a competing interest but thought I should let BMJ know.
REVIEW RETURNED	03-Jan-2017

GENERAL COMMENTS	Overall This is a well-written manuscript on a highly important topic. Most of my recommendations/comments are to improve readability/clarity. My greatest concern is the first bullet under ‘Discussion’ below. STROBE  • Title: Recommend adding the term ‘cross-sectional analysis’ to title • Descriptive data: Most tables list missing data, but there is no indication under ‘Obstetric emergency’ for Table 4. Does every transport have a diagnosis (some obstetric emergency v normal labor)? General formatting:  • The term ‘inter-facility transfer’ is abbreviated to IFT midway through manuscript (page ***). Recommend doing this from initial mention in the Introduction. • Although this manuscript contains many interesting findings, it does seem a bit long. If there were need to trim down, I would consider removing the content regarding the timing of IFT and non-
--

IFT calls. It does not seem to add to the overall thrust of the paper.

- Please be consistent in using either “destination hospital” or “referral hospital”.

Abstract

- Recommend spelling out “inter-facility transfer” once at beginning of abstract for less familiar readers.
- Line 31-32: recommend using your term IFT instead of “transferred between facilities” will better stand out to readers as your primary outcome. It’s the same thing, just a readability.
- Results: You include the OR for IFT when an obstetric emergency is reported, however you comment on other associations. Is there room to report the other OR’s – particularly urban area and BPL status (seem to be slightly better predictors than others listed)?

Strengths & Limitations

- Final bullet: consider adding how you “appropriately dealt” with the missing data. I think this can be done briefly by stating that you did a complete cases analysis and sensitivity analysis.

Introduction

- Very strong. Does well at setting the stage.
- For the paragraph on page 7 starting at line 18, may be able to remove description of Assam – it is addressed more directly later in discussion. It seems a bit out of place amongst the more general description of types services (public, private, JEY, etc).

Methods

- Context: Page 8, first sentence, first paragraph: the reference cited states that ‘108’/‘102’ is available in 31 states/union territories, rather than 21. Is the difference just states in which only ‘102’ exists?
- Study population: the inclusion criteria include “pregnant women who called... or for home a friend or relative called.” The criteria do not seem to included health care providers. I imagine that often, although potentially not always, IFTs are initiated by health care providers (doctors, nurses, or midwives). How were these calls found in the database?
- Study population: did this study population include all pregnant women, regardless of trimester and including postpartum?
 - o Are there any limitations to your methods as far as missing pregnant women who called for what may have been considered a non-obstetric emergency?
- Obtaining data: “Any gross issue relating to the quality of records was noted...” Is the missing data the only ‘gross issue’ that arose? Were there others? And if so, how were these addressed. If not, you may want to consider removing this sentence.
- Obtaining data: “Variables were recoded wherever needed.” What does this relate to? I’m not sure if this adds clarity or raises an unnecessary question in the mind of the reader.
- Analysis: Is there a reference you can include regarding your 1.1 multiplier and the formulas you use?
- Analysis: Thank you for describing your case analysis and your sensitivity analysis – very helpful. Can you clarify what you were looking for as far a “substantial change in the magnitude of the pattern”?

Results

- Table 1: What is the variable ‘Region’? I don’t believe it is mentioned elsewhere in the manuscript.
- Of the 608,559 pregnant women, how many were postpartum?

	 • In addition, to reporting that 0.15% of pregnant women died prior to hospital arrival in Chhattisgarh, might it be helpful to report it as a rate, similar to the sentence before? • Table 3: You report the percent of total that IFTs make up, please include similar % for non-IFTs. • Table 3: Is there a different way to present the % missing? Perhaps as a footnote to the table? • Page 18, second line of first paragraph: “overall total” should be “overall average” • Table 4: I noticed that sepsis is not one of the listed obstetric emergencies. Is this included under ‘medical condition’? Or is it considered an ‘other emergency’? How were these categories chosen? This should be addressed in your section “Working definitions” under Methods.  o Similarly, where would other not listed pregnancy related emergencies be listed (e.g. pulmonary embolism, amniotic fluid embolism, HELLP, severe preeclampsia)? • Table 4: Is there space to include (as footnote or otherwise) what the hospital acronyms stand for and their approximate level (i.e. primary, secondary, tertiary)? This may help readers better analogize your findings to their own setting. • Figure 1 needs a title. Discussion  • What proportion of institutional deliveries would you want transported by ‘108’? Are there recommendations? • What proportion of deliveries do you anticipate require an interfacility transfer? Discussion  • You conclude that “the ‘108’ service is not a preferred choice for transport to a higher-level facility, or for obstetric emergencies.” You base this conclusion off the percent of total institutional deliveries transferred by ‘108’ to a higher level facility. Do you know the total number of institutional deliveries in these states during your study period that required and request transfer to higher level facilities? This would seem to be the required denominator to make this conclusion. That is, if health care providers only sought to transfer 40,000 women to higher level facilities during this period and ‘108’ transferred approximately 34,993 women, then they would be the preferred method and the root problem would be failure to transfer. In fact, in the following paragraph you estimate that between 7% to 19% of all non-IFTs transferred by 108 may require further transfer, which is 40,149 to 80,299 women. Consequently, ‘108’ is transferring between 44% to 87% of those non-IFTs that you estimated would require further transfer. Please address and/or clarify. • You mention that ‘108’ doctors were often not convinced of the need for IFT. In those cases, how is the need for IFT coded? That is, was it that 92% of the time referring hospitals perceived no emergency or complication, or that 92% of the time it was the ‘108’ doctors conclusion that there was no emergency or complication.
--	--

REVIEWER	Sarika Chaturvedi Savitribai Phule Pune University
REVIEW RETURNED	14-Jan-2017

GENERAL COMMENTS

This is an interesting paper presenting results of a very relevant study.

I have the following comments on the manuscript:

1. Use of acronyms or short forms in abstract is better avoided. Pl spell IFT and nonIFT in the abstract.
2. Authors may recheck use of the term 'cross sectional analysis of ambulance records'. 'A retrospective review of ambulance records' would be better instead.
3. State in India is conventionally spelled as 'state' and not with 'S', pl correct throughout.
4. Page 7- Line 18-20: The authors mention India has no structured inter facility referral and transportation protocols. Could the authors clarify in their response the basis for this. In case the authors intend to write about the possible gaps between existing protocols and routine practices in public facilities, it should be specified so.
5. Page 8- Lines 14-27 Objectives: I suggest the authors restate their objectives to make it easier like : (i) uptake/ usage (demand) of 108 in among institutional deliveries in study population (iii) proportion of IFTs among institutional deliveries (iii) use of 108 in IFTs among institutional deliveries and (iv) comparison of characteristics of women using 108 for IFT and non IFT
6. Page 8 Methods- In the description of the ambulance, the authors mention as ' EMT should...'; it would be better to either clarify this as 'is expected to' or 'GVK ensures EMTs do.....'.
7. In order to simplify, the authors may reword 'interfacility transfer' as referral and 'non IFT' as pickups for institutional delivery as is understood from the definition.
8. It would be interesting if the authors present the number/proportion of referrals made by hospitals in the study period, thus allowing to assess the contribution of '108' to overall referrals/IFTs. This would be more interest from a health systems perspective than a mere description of 108 usage.
9. The authors need to clarify about the data on obstetric emergencies- how obstetric emergencies were identified and by whom. This information is important to interpret the data. Although this is mentioned briefly in the limitations, it would be useful to describe this in the methods.
10. The authors may justify their age classification; it is not clear why 5 age groups have been made , it would be better to make a logical classification. Although the results might not change, it might help

comprehending the data better.

11. Page 18- Obstetric emergencies: Authors mention of more obstetric emergencies among IFTs, which is obvious. It would be useful to know the authors' reason in describing this.

12. Page 18- Destination facilities: The authors mention in majority cases transfers were to government facilities. It would be useful to clarify in the context/background the availability of non government sector in the study states and whether '108' system has any specific protocol in choosing between government and other facilities or do the users have a choice.

13. Page 19- Distance and time travelled- It would be useful if the authors clarify in the methods how distance and travel time is recorded in GVK records; whether it is as reported by the drivers/users or GPS based.

14. Page 19- Distance and time travelled- For about half of IFTs the distance was less than 4 Km. Could the authors reconfirm this data; given the geographic spread of facilities and especially government facilities 4Km appears to be much underestimated. Similarly for the IFT time. Further could the authors explain why the time required for lesser median distance in Gujarat is relatively longer(as seen from Table 4).

The authors may compare these time/distance finding with other published data from Indian states.

15. Table 4- Obstetric emergencies- Previous caesarean and precious pregnancy are not emergencies per say. It will be better the authors keep terminology consistent with accepted international usage.

16. Table 4- Title can be more concise, 'different states' seems obsolete. Mentioning an overall N in the title would be useful.

17. As for age classification, over categorisation adds no value to facility/destination. The authors may merge appropriately for better comprehension.

18. The presentation of data for different district and different mandal is not clear- why do the authors think this is important to be presented?

19. Figure 1 needs a legend.

20. In the discussion, (i) the authors may allude to previous reports on GVK-EMRI usage in India (even if the reports are in-house reports) and the effect on existing public referral/transport system. (ii) It would be important that the authors mention and discuss literature on different models of referral and transport for obstetric emergencies and in this light what do the authors recommend would be the lessons for low income contexts. (iii) I find the cost factors have not been considered in the paper, it would be meaningful to

	consider these in the background and the discussion- what is known of government expenditures on referral transport/GVK-EMRI model, how does this compare with other models in India such as the Janani Express; the authors recommendations need to be substantiated with these considerations. (iv) Another important aspect the results reveal is the GVK-EMRI being seen as an easy way to pass off patients to higher centres; this is likely to overload the already burdened referral facilities when lower level facilities are less equipped. The authors need to bring this more clearly in the discussion also referring to the importance of competent and accountable providers for proper functioning of a referral chain.
--	---

VERSION 1 – AUTHOR RESPONSE

Reviewer: 1

Reviewer Name: Lee Wallis

An interesting and well written paper covering an important topic in MCH

Thank you

IFT is spelled out multiple times before being abbreviated

Response: Thank you for pointing out.

Edit: Corrected at several places.

My main issues are twofold.

The first is length: i am concerned it is very long for what is basically a report of EMS case types (4800+ words) It could easily be cut down significantly

Response: We welcome your suggestion and have shortened it as much as we could. Other reviewers wanted more elaboration at some places.

Edit: At several places.

The second lies in the methods: there is no justification for the dates chosen, and little detail about how and why the 5 states were chosen (other than that they had EMRI for a seemingly arbitrary 3 years). does this mean that all other states had EMRI for less time? does that matter? why 3 years?

Response: '108' systems in India were adopted by states over 4-7 years at different time points, prior to the study period. Within states the ambulances were deployed in phased manner and '108' system reached to its full capacity by 2-3 years. Thus we decided an eligibility criteria of three years since launch of '108' systems in the respective states. We chose one year data for the period (1st April – 31st March) such that it is aligned with reporting period for the state governments and could be used for triangulation with other information. About selecting five states, we chose from list of 10 eligible states such that we had representation from North, South, East, and West of the country. We have added about this briefly in the text.

Edit: Page 10, Line 1-3

Page 11 line 21-23: how valid is this 10% multiplier assumption? line 35-37: how valid is this assumption?

otherwise i am happy with the paper

Response: It is a hypothetical percentage adopted by Government of India to estimate pregnancies

which may have ended in abortions or intra-uterine deaths. We have added the reference for this. We accept that the validity of this percentage is doubtful. We have added this in limitations.

Edit: Page 5, Line 13; Page 11, Line 13

Reviewer: 2

Reviewer Name: Wally Carlo

This is an important manuscript as it reports a cross sectional analysis of a large emergency maternal ambulance service in large populations.

Title

“108” is not a universally-used concept. Maybe a descriptor that is of more general use such as “emergency free ambulances” or “emergency call center ambulances” could be used instead in the title and text.

Response: We have edited the title as per your suggestion. In India there are a few publically funded or subsidized ambulance services as described in the Introduction para 4. The ‘108’ service is the largest and we have analysed specifically the ‘108’ services operated by GVK-EMRI thus we believe we shall retain this in results and discussion.

Edit: Title

Abstract

Include the % rather than just interpret as “very small” the proportion that used the ‘108’ ambulance for transfer between facilities. This is also pertinent to the last paragraph of the manuscript.

Response: Added.

Edit: Page 3, Line 21

Background

The Background section includes details that are not directly related to the research presented and that are less generalizable to other settings such as alternative transport systems in various Indian states. Thus, the 4th paragraph should be shortened substantially.

Response: Edited as suggested

Edit: Page 7, Line 6-17

Methods

Whenever possible, generic names and not commercial names, such as GVK-EMRI, should be used.

Response: ‘108’ service is operated in 21 states and union territories of which GVK-EMRI operates the ‘108’ service in 17 (mentioned in Context). In our study we analysed data only from states where GVK-EMRI operated. We have mentioned GVK-EMRI only where absolutely necessary.

It would be good to have some explanation of the classification of castes for an international journal.

Response: We have added the information in the text.

Edit: Page 10, Line 13,16

Results

Table 1 should include units for all variables (including MMR and NMR) not just for some.

Edited table-1.

Data on the time of days of the transports is commented on in the Discussion but should be presented in Results.

Response: Already presented in the sub-section Time of day and day of the week in results. However to reduce the size of the paper and as suggested by other reviewer we have deleted whole section.

Many abbreviations are not spelled out.

Response: Thank you for pointing out. Spelled out in the text wherever required.

Reviewer: 3

Reviewer Name: Jennifer A. Newberry

Overall

This is a well-written manuscript on a highly important topic. Most of my recommendations/comments are to improve readability/clarity. My greatest concern is the first bullet under 'Discussion' below.

Thank you

We observe that you have several suggestion about obstetric emergencies. This paper is dedicated to IFTs and obstetric emergencies are important determinant here. We have described obstetric emergencies in detail in a dedicated paper published elsewhere. We have in brief addressed your concerns in this paper and mentioned the reference to the other paper wherever details are required.

STROBE

- Title: Recommend adding the term 'cross-sectional analysis' to title

Edit: Added in the Title.

- Descriptive data: Most tables list missing data, but there is no indication under 'Obstetric emergency' for Table 4. Does every transport have a diagnosis (some obstetric emergency v normal labor)?

Response: There was no missing data under this variable. We recoded the variable named type of emergency to generate Obstetric emergency v normal labour/others. We have described this in detail in other research paper specifically on Obstetric emergencies from same data set. We have added these in brief in this paper and added reference for the other paper.

Edit: Page 10, Line 9

General formatting:

- The term 'inter-facility transfer' is abbreviated to IFT midway through manuscript (page ***).

Recommend doing this from initial mention in the Introduction.

Response: Thank you for pointing out.

Edit: Corrected at several places.

- Although this manuscript contains many interesting findings, it does seem a bit long. If there were need to trim down, I would consider removing the content regarding the timing of IFT and non-IFT calls. It does not seem to add to the overall thrust of the paper.

Response: We welcome your suggestion and have shortened it as much we could.

Edit: at several places

- Please be consistent in using either "destination hospital" or "referral hospital".

Response: A 'destination hospital' for non-IFT can also be a PHC which is not considered a referral hospital. Thus we used the term destination hospital in results. We use referral facility only when describing IFTs to build a context in introduction and discussion.

Edit: Page 19, Line 15

Abstract

- Recommend spelling out "inter-facility transfer" once at beginning of abstract for less familiar readers.

Response: Spelled out.

Edit: Page 3, Line 2

- Line 31-32: recommend using your term IFT instead of "transferred between facilities" will better stand out to readers as your primary outcome. It's the same thing, just a readability.

Response: Thank you.

Edit: Page 3, Line 11, 19

- Results: You include the OR for IFT when an obstetric emergency is reported, however you comment on other associations. Is there room to report the other OR's – particularly urban area and BPL status (seem to be slightly better predictors than others listed)?

Response: We chose to mention for obstetric emergencies as this was our main determinant under study. Adding OR for others will increase the word count above that acceptable for abstract.

Edit: None

Strengths & Limitations

- Final bullet: consider adding how you “appropriately dealt” with the missing data. I think this can be done briefly by stating that you did a complete cases analysis and sensitivity analysis.

Response: Thank you for the suggestion. Added.

Introduction

- Very strong. Does well at setting the stage. Thank you
- For the paragraph on page 7 starting at line 18, may be able to remove description of Assam – it is addressed more directly later in discussion. It seems a bit out of place amongst the more general description of types services (public, private, JEY, etc).

Response: Deleted as suggested.

Methods

- Context: Page 8, first sentence, first paragraph: the reference cited states that ‘108’/‘102’ is available in 31 states/union territories, rather than 21. Is the difference just states in which only ‘102’ exists?

Response: Yes there are states that have only ‘102’ ambulances.

- Study population: the inclusion criteria include “pregnant women who called... or for home a friend or relative called.” The criteria do not seem to include health care providers. I imagine that often, although potentially not always, IFTs are initiated by health care providers (doctors, nurses, or midwives). How were these calls found in the database?

Response: Thank you. We actually included calls by health care providers as well. We have included this now.

Edit: Page 9, Line 20

- Study population: did this study population include all pregnant women, regardless of trimester and including postpartum?

Response: Yes it included all women irrespective of trimester. It may also have included post-partum women with complications but we could not extract these with given data. But we could determine post-partum women who were transferred back from hospital to home post-delivery. We excluded these women from analysis.

- Are there any limitations to your methods as far as missing pregnant women who called for what may have been considered a non-obstetric emergency?

Response: We suppose we did not miss any pregnant women. Any pregnant women calling for ambulance was recorded as pregnancy related call. Within this category, for type of emergency, few emergencies such as suicides, accidents and some medical ailments were also mentioned. These other emergencies were reported in less than 2%. This is described in detail in other published paper dedicated on obstetric emergencies. We have added the reference.

Edit: Page 10, Line 9

- Obtaining data: “Any gross issue relating to the quality of records was noted...” Is the missing data the only ‘gross issue’ that arose? Were there others? And if so, how were these addressed. If not, you

may want to consider removing this sentence.

Response: We have briefly mentioned about the management of data in this paper. Details are described in other paper. We have added the reference to this section.

Edit: Page 10, Line 9

- Obtaining data: "Variables were recoded wherever needed." What does this relate to? I'm not sure if this adds clarity or raises an unnecessary question in the mind of the reader.

Response: We recoded for variables such as age and obstetric emergency. The management of data is described in detail in another paper. We have added the reference to this section.

Edit: Page 10, Line 9

- Analysis: Is there a reference you can include regarding your 1.1 multiplier and the formulas you use?

Response: Reference added.

Edit: Page 11, Line 13

- Analysis: Thank you for describing your case analysis and your sensitivity analysis – very helpful. Can you clarify what you were looking for as far as a "substantial change in the magnitude of the pattern"?

Response: We looked for ORs for each of the variables and Rsq of the models. Specified in text.

Edit: Page 12, Line 14

Results

- Table 1: What is the variable 'Region'? I don't believe it is mentioned elsewhere in the manuscript.

Response: This refers to geographic terrain to provide context for states. In the study we have only studied area (rural/tribal and urban).

Edit: Table 1

- Of the 608,559 pregnant women, how many were postpartum?

Response: We excluded post-partum women who were transferred back from hospital to home. Other post-partum women transported for any complication could not be ascertained from the data. Added in limitations

Edit: Page 5, Line 8-9

- In addition, to reporting that 0.15% of pregnant women died prior to hospital arrival in Chhattisgarh, might it be helpful to report it as a rate, similar to the sentence before?

Response: Thank you for the suggestion

Edit: Page 15, Line 13

- Table 3: You report the percent of total that IFTs make up, please include similar % for non-IFTs.

Response: Added

- Table 3: Is there a different way to present the % missing? Perhaps as a footnote to the table?

Response: Thank you for the suggestion. There are missing in almost each variable in each state for both IFT and non-IFT, and is difficult to add all of these as footnote. We deliberated on this a lot, but found the current presentation more appropriate.

Edit: None

- Page 18, second line of first paragraph: "overall total" should be "overall average"

Response: Thank you for the suggestion

Edit: Page 17, Line 3

- Table 4: I noticed that sepsis is not one of the listed obstetric emergencies. Is this included under

'medical condition'? Or is it considered an 'other emergency'? How were these categories chosen? This should be addressed in your section "Working definitions" under Methods.

- Similarly, where would other not listed pregnancy related emergencies be listed (e.g. pulmonary embolism, amniotic fluid embolism, HELLP, severe preeclampsia)?

Response: We used the categories recorded in the database. The obstetric emergencies were recorded as mentioned by the caller or the EMT. More complicated diagnosis are likely to be recorded as others. We have described this in detail in other research paper. We have added the reference in the methods.

Edit: Page 9, Line 7-12; Page 10, Line 9

- Table 4: Is there space to include (as footnote or otherwise) what the hospital acronyms stand for and their approximate level (i.e. primary, secondary, tertiary)? This may help readers better analogize your findings to their own setting.

Response: Thank you for the suggestion. Added in table.

Edit: Table 4

- Figure 1 needs a title.

Response: Thank you for pointing out. We have deleted this section from the paper to reduce size.

Edit: Figure-1 deleted.

Discussion

- What proportion of institutional deliveries would you want transported by '108'? Are there recommendations?

- What proportion of deliveries do you anticipate require an interfacility transfer?

Response: There are no such recommendations. Government wants to provide ambulance for all those who need it and do not have access to other transport/ambulance.

A review from India suggest that 14-36% women visiting lower levels of care or BEmOC for delivery are referred and would require transport. A study found 15% of all institutional deliveries in their study districts were IFTs. We have discussed this in discussion section.

Discussion

• You conclude that "the '108' service is not a preferred choice for transport to a higher-level facility, or for obstetric emergencies." You base this conclusion off the percent of total institutional deliveries transferred by '108' to a higher level facility. Do you know the total number of institutional deliveries in these states during your study period that required and request transfer to higher level facilities? This would seem to be the required denominator to make this conclusion. That is, if health care providers only sought to transfer 40,000 women to higher level facilities during this period and '108' transferred approximately 34,993 women, then they would be the preferred method and the root problem would be failure to transfer. In fact, in the following paragraph you estimate that between 7% to 19% of all non-IFTs transferred by 108 may require further transfer, which is 40,149 to 80,299 women. Consequently, '108' is transferring between 44% to 87% of those non-IFTs that you estimated would require further transfer. Please address and/or clarify.

Response: Thank you for raising an important point. We found that only 1% of all institutional deliveries in the population were transported by '108' for IFT. There is no estimate for the number of referrals in the states for the study period. But as mentioned above, evidence from India suggests that 14-36% of institutional deliveries are referred from one facility to other. Considering two things, a) the pattern of use of health facilities by non-IFTs by '108' and b) the pattern of referrals from review of evidence from India, we estimated that 7% to 19% (which is 40,000 to 80,000 women) of all non-IFTs transferred by '108' may further require IFT. In addition, among the estimated 80% institutional deliveries (2,300,000) who used other means of transport (non-108), some women may be referred further. We have clarified this in the text.

Edit: Page 24, Line 1-8

• You mention that '108' doctors were often not convinced of the need for IFT. In those cases, how is the need for IFT coded? That is, was it that 92% of the time referring hospitals perceived no emergency or complication, or that 92% of the time it was the '108' doctors conclusion that there was no emergency or complication.

Response: We would like to clarify, that 92% of the time the database has recorded the cause of call as normal labour or other non-obstetric cause (less than 2%). Mostly it was the '108' doctors conclusion after speaking with the referring health care provider (caller) that there was no obstetric emergency or complication. The diagnosis was reported by EMT after telephonic consultation with '108' doctor. We have clarified this better in the methods section and discussed later.

Edit: Page 9, Line 7-12

Reviewer: 4

Reviewer Name: Sarika Chaturvedi

Thank you for your review

We observe that you have several suggestion about obstetric emergencies. This paper is dedicated to IFTs and obstetric emergencies are important determinant here. We have described obstetric emergencies in detail in a dedicated paper published elsewhere. We have in brief addressed your concerns in this paper and mentioned the reference to the other paper wherever details are required.

1. Use of acronyms or short forms in abstract is better avoided. Pl spell IFT and nonIFT in the abstract.

Response: We accept the point made. However spelling out IFT and non-IFT throughout will add many additional words and we will have to delete some important information to stick to word limit. We have spelled IFT and abbreviated it at first use in abstract. If not mandatory, we will like to keep it this way.

2. Authors may recheck use of the term 'cross sectional analysis of ambulance records'. 'A retrospective review of ambulance records' would be better instead.

Response: Thank you for your suggestion. We would like to retain cross-sectional analysis of ambulance records.

3. State in India is conventionally spelled as 'state' and not with 'S', pl correct throughout.

Response: Corrected at various places

4. Page 7- Line 18-20: The authors mention India has no structured inter facility referral and transportation protocols. Could the authors clarify in their response the basis for this. In case the authors intend to write about the possible gaps between existing protocols and routine practices in public facilities, it should be specified so.

Response: This is based on extensive desk review and personal communication by Maternal health specialist in Government of India. Added in text.

Edit: Page 7; Line 6-8

5. Page 8- Lines 14-27 Objectives: I suggest the authors restate their objectives to make it easier like : (i) uptake/ usage (demand) of 108 in among institutional deliveries in study population (ii) proportion of IFTs among institutional deliveries (iii) use of 108 in IFTs among institutional deliveries and (iv) comparison of characteristics of women using 108 for IFT and non IFT

Response: Thank you for simplifying. We have edited these.

Edit: Page 7, Line 22 to Page 8, Line 4

6. Page 8 Methods- In the description of the ambulance, the authors mention as ' EMT should...'; it would be better to either clarify this as 'is expected to' or 'GVK ensures EMTs do.....'.

Response: Thank you. We have edited this to 'is expected to'.
Edit: Page 8 Line 16.

7. In order to simplify, the authors may reword 'interfacility transfer' as referral and 'non IFT' as pickups for institutional delivery as is understood from the definition.

Response: Thank you for the suggestion. We reviewed the literature and found that term 'referral' is used for both self-referral and referral by health worker. As we were assessing the referral transport for inter-facility transfer we deliberately chose the term inter-facility transfer. Some non-IFTs were also for complications during antenatal period and postnatal period. Thus we prefer term non-IFTs over pick-ups for institutional delivery.

Edit: None

8. It would be interesting if the authors present the number/proportion of referrals made by hospitals in the study period, thus allowing to assess the contribution of '108' to overall referrals/IFTs. This would be more interest from a health systems perspective than a mere description of 108 usage.

Response: Unfortunately such information was not available for the study period. However our systematic review from India found that 14-36% pregnant women visiting health facility for delivery may require IFTs. This is described in discussion.

Edit: None

9. The authors need to clarify about the data on obstetric emergencies- how obstetric emergencies were identified and by whom. This information is important to interpret the data. Although this is mentioned briefly in the limitations, it would be useful to describe this in the methods.

Response: We have added this in brief in methods and also added reference to our other paper.

Edit: Edit: Page 9, Line 7-12

10. The authors may justify their age classification; it is not clear why 5 age groups have been made, it would be better to make a logical classification. Although the results might not change, it might help comprehending the data better.

Response: Age <19 years and >35 years are known to have higher probability of having complications so we specified these groups. To observe trends with age we divided the 20-34 years group into 20-24years, 25-29years and 30-34years. Same distribution is also used in other national surveys.

Edit: None.

11. Page 18- Obstetric emergencies: Authors mention of more obstetric emergencies among IFTs, which is obvious. It would be useful to know the authors' reason in describing this.

Response: Although obstetric emergencies were more in IFTs (8.4%) they were not much higher than non-IFTs (4.4%). 91% of IFTs were for non-obstetric emergency conditions. We have discussed this in discussion section.

Edit: None

12. Page 18- Destination facilities: The authors mention in majority cases transfers were to government facilities. It would be useful to clarify in the context/background the availability of non-government sector in the study states and whether '108' system has any specific protocol in choosing between government and other facilities or do the users have a choice.

Response: '108' system preferred to transfer women to public facility. We have added this in context in methods.

Edit: Page 8, Line 20-22.

13. Page 19- Distance and time travelled- It would be useful if the authors clarify in the methods how distance and travel time is recorded in GVK records; whether it is as reported by the drivers/users or GPS based.

Response: EMTs reported ambulance meter readings at start of end of the trip. We have added this in methods.

Edit: Page 10, Line 17

14. Page 19- Distance and time travelled- For about half of IFTs the distance was less than 4 Km. Could the authors reconfirm this data; given the geographic spread of facilities and especially government facilities 4Km appears to be much underestimated. Similarly for the IFT time. Further could the authors explain why the time required for lesser median distance in Gujarat is relatively longer(as seen from Table 4).The authors may compare these time/distance finding with other published data from Indian states.

Response: '108' ambulances are stationed in or near the health facilities. Thus the distance and time to reach pick-up hospital for IFT patients is short. But the distance and time to reach destination is longer in IFTs compared to non-IFTs. We have discussed this in the paper. About Gujarat, we cannot determine the reason from the secondary data.

Edit: None

15. Table 4- Obstetric emergencies- Previous caesarean and previous pregnancy are not emergencies per say. It will be better the authors keep terminology consistent with accepted international usage.

Response: This is rightly pointed by you. We observed that '108' services treat them as emergencies. Women in labour with previous caesarean and previous pregnancy when in transit may require more aggressive management in case of imminent childbirth. We have described this under definition.

Edit: Edit: Page 9, Line 7-12

16. Table 4- Title can be more concise, 'different states' seems obsolete. Mentioning an overall N in the title would be useful.

Edit: Table 4 title.

17. As for age classification, over categorisation adds no value to facility/destination. The authors may merge appropriately for better comprehension.

Response: We wanted to assess any trend in proportion of IFTs with respect to age group thus this categorization was done. Justification of the categories chosen is mentioned above.

Edit: None.

18. The presentation of data for different district and different mandal is not clear- why do the authors think this is important to be presented?

Reference: We wanted to study if the referral facility was within the same administrative unit or other. Continuity of care and monitoring of IFTs can be better if transfers are in same administrative unit. Secondly, transfer to different district suggests the sparse distribution of referral facilities. We have added this in discussion.

Edit: Page Line

19. Figure 1 needs a legend.

Reference: Thank you for reminding. But we have deleted this section to reduce the size of the paper as suggested by other reviewer.

20. In the discussion,

(i) the authors may allude to previous reports on GVK-EMRI usage in India (even if the reports are in-house reports) and the effect on existing public referral/transport system.

Reference: As suggested we assessed previous reports but there is not much information on IFTs for pregnant women. There is some mention of obstetric emergencies which we discussed in our other published paper on transport for pregnant women and obstetric emergencies.

Edit: None.

(ii) It would be important that the authors mention and discuss literature on different models of referral and transport for obstetric emergencies and in this light what do the authors recommend would be the lessons for low income contexts.

Reference: Your suggestion is valid. We published a separate paper on transport for pregnant women and obstetric emergencies. There we discussed other models too. Present paper focusses on IFT, and as per your suggestion we have added comparison with other models with relevance to IFT.

Edit: Page 26, Line 13-22

(iii) I find the cost factors have not been considered in the paper, it would be meaningful to consider these in the background and the discussion- what is known of government expenditures on referral transport/GVK-EMRI model, how does this compare with other models in India such as the Janani Express; the authors recommendations need to be substantiated with these considerations.

Reference: Estimation of costs was not in scope of this paper. We looked at literature and found that cost estimates were not stratified for pregnant women or IFT for pregnant women among all general transfers by '108'.

Edit: None

(iv) Another important aspect the results reveal is the GVK-EMRI being seen as an easy way to pass off patients to higher centres; this is likely to overload the already burdened referral facilities when lower level facilities are less equipped. The authors need to bring this more clearly in the discussion also referring to the importance of competent and accountable providers for proper functioning of a referral chain.

Reference: Thank you for your suggestion. We have added this.

Edit: Page 25, Line 7-9

VERSION 2 – REVIEW

REVIEWER	Lee Wallis University of Cape Town South Africa
REVIEW RETURNED	27-Feb-2017

GENERAL COMMENTS	much better thanks
--------------------

REVIEWER	Waldemar Carlo University of Alabama at Birmingham
REVIEW RETURNED	15-Mar-2017

GENERAL COMMENTS	The authors have responded well to my comments.
---

VERSION 2 – AUTHOR RESPONSE

Reviewer: 1

Reviewer Name: Lee Wallis

much better thanks

Thank you

Reviewer: 2

Reviewer Name: Waldemar Carlo

The authors have responded well to my comments.
Thank you